# Preventive and Therapeutic Efficacy of Roselle Beverage Residue in Late-Stage Type 2 Diabetic Rats

**Evelyn Regalado-Rentería** [1,†] , **Jesús E. Serna-Tenorio** [2,†] , **David G. García-Gutiérrez** [1] , **Rosalía Reynoso-Camacho** [1] , **Olga P. García** [2] , **Miriam A. Anaya-Loyola** [2] **and Iza F. Pérez-Ramírez** [1,*]

1   Facultad de Química, Universidad Autónoma de Querétaro, Querétaro 76010, Mexico;
    evelyn.regalado.renteria@gmail.com (E.R.-R.); davidggtz@gmail.com (D.G.G.-G.);
    rrcamachomx@yahoo.com.mx (R.R.-C.)
2   Facultad de Ciencias Naturales, Universidad Autónoma de Querétaro, Querétaro 76010, Mexico;
    eduardoserna1808@gmail.com (J.E.S.-T.); olga.garcia@uaq.mx (O.P.G.); aracely.anaya@uaq.mx (M.A.A.-L.)
*   Correspondence: iza.perez@uaq.mx
†   These authors contributed equally to this work.

**Abstract:** The residue from roselle beverage production is rich in polyphenols and dietary fiber. We investigated its potential as a preventive and therapeutic agent for type 2 diabetes mellitus (T2DM). Male Wistar rats were fed a high-fat high-fructose diet (HFFD) for 17 weeks, reaching insulin resistance by week 9, and induced to T2DM with streptozotocin (STZ) at week 13. Roselle beverage residue (RBR) was administered ad libitum mixed at 6% with the HFFD. Rats received HFFD+RBR as a preventive strategy starting at week 1 (healthy) and week 9 (insulin resistant), whereas the treatment strategy in T2DM rats started at week 14 alone or in combination with metformin (200 mg/kg/day), with a control metformin-treated group. All RBR-supplemented groups showed reduced serum glucose levels (1.4-fold to 1.8-fold) compared with the HFFD+STZ control group. Preventive RBR administration enhanced pancreatic function, leading to improved insulin sensitivity (6.5-fold to 7.9-fold). Gene expression analysis identified slight alterations in hepatic and skeletal muscle glucose metabolism. Additionally, RBR supplementation demonstrated a preventive role in mitigating hyperuricemia (2.1-fold to 2.2-fold), with no effect on glomerular hyperfiltration. While the exact mechanisms underlying RBR effects remain to be fully elucidated, our findings highlight its promising potential as a dietary supplement for preventing and treating T2DM.

**Keywords:** roselle; *Hibiscus sabdariffa* L.; residue; by-product; insulin resistance; type 2 diabetes mellitus; rats; nephropathy



## 1. Introduction

Type 2 diabetes mellitus (T2DM) continues to be a global health concern, with one of ten adults worldwide living with this condition, and a significant proportion remaining undiagnosed. The 2023 campaign of the International Diabetes Federation (IDF) emphasized the importance of timely care for effective treatment and management strategies [1]. The primary approach to T2DM prevention and control involves the adoption of healthy lifestyle habits. For instance, the Diabetes Prevention Program (DPP) of the American Diabetes Association (ADA) includes an exercise intervention as well as modifications in dietary patterns; the program targets prediabetic individuals with established insulin resistance to prevent the development of T2DM [2]. The goal of nutritional therapy is to achieve normoglycemia and insulin sensitivity, with dietary fiber-rich foods as a dietary strategy [3,4].

In addition, the ADA acknowledges that certain dietary supplements may offer potential benefits for individuals with T2DM or risk factors, although further research is required to establish the efficacy of plant-based supplements in improving glycemic control [3]. It is

important to consider that these nutritional therapy strategies do not substitute pharmacological interventions such as metformin, an oral hypoglycemic drug often prescribed for those at risk of vascular diseases associated with T2DM [3].

Several polyphenols have shown potential in preventing, managing, and even reversing diabetes. These bioactive compounds, found in numerous dietary sources, offer a multi-target approach to address the multifactorial nature of this disease [5,6]. Furthermore, studies suggest that even low doses of certain polyphenols, like gallic acid, can enhance the antidiabetic effects of antihyperglycemic drugs like metformin [7].

The calyxes of roselle (*Hibiscus sabdariffa* L.) have been identified as a rich source of phenolic compounds with antioxidant and hypoglycemic potential, supported by in vitro assays [8]. Interestingly, some roselle polyphenols, including quercetin, hibiscetin, and protocatechuic acid, have potential in inhibiting phosphoenolpyruvate carboxykinase (PEPCK), an enzyme involved in hepatic gluconeogenesis, suggesting a role in glycemic control [9]. Additionally, polyphenol-rich extracts of roselle calyxes promoted insulin secretion [10], decreased hyperglycemia [11], and improved aortic oxidative stress [12] in type 1 diabetic rats. Despite these findings, no prior research has evaluated the effect of roselle calyxes in glucose management in the context of T2DM.

Our prior study demonstrated that powdered roselle calyx improved insulin resistance in obese rats fed a hypercaloric diet, an effect also observed with the residue from roselle decoction [12]. This residue is produced in significant quantities during the preparation of roselle-based beverages in home settings and food processing industries. In comparison with roselle calyxes, the residue has a higher content of dietary fiber and non-extractable polyphenol, but lower levels of extractable polyphenols [13,14]. Nevertheless, the effect of this residue has not been assessed for its preventive and therapeutic effect on T2DM, which could contribute to reducing waste and promoting sustainability. Moreover, this residue could offer an accessible and cost-effective nutraceutical intervention for individuals at risk or diagnosed with T2DM, particularly in regions were roselle-based beverages are commonly consumed.

Huang et al. [15] demonstrated that supplementation with a tea residue rich in dietary fiber and non-extractable polyphenols effectively prevented T2DM complications in a rodent model. Therefore, we hypothesized that roselle beverage residue (RBR), due to its dietary fiber and non-extractable polyphenol content and composition, can serve as a dietary supplement to prevent and manage hyperglycemia in a T2DM rat model. Specifically, this study aims to assess the impact of RBR on the expression of genes involved in insulin resistance in T2DM-induced rats.

## 2. Materials and Methods

### 2.1. Preparation of Roselle Beverage Residue (RBR)

Roselle (*Hibiscus sabdariffa* L.) calyxes were obtained from local farmers in Guerrero, Mexico. The calyxes underwent a disinfection process via immersion in Nobac® citrus 373 solution (1% $v/v$) for 10 min, followed by drying at 45 °C for 24 h in a forced circulation oven (BF 400, Binder GmbH, Tuttlingen, Germany), as previously described [13]. Subsequently, the dried and disinfected calyxes (100 g) were subjected to decoction with boiling water (1000 mL) for 15 min. The resulting cooked calyxes (RBR) were recovered, dried, ground to <420 nm particle size, and stored at −20 °C in light-protected, hermetically sealed containers until further use.

### 2.2. In Vivo Experimental Design

Seventy male Wistar rats (180–200 g) were obtained from the Neurobiology Institute at the Universidad Nacional Autónoma de Mexico, Querétaro, Mexico. Rats were individually housed under controlled conditions of temperature (20–25 °C), relative humidity (30–60%), and a 12 h light-dark cycle with continuous ventilation. All experimental procedures were conducted following the animal care guidelines (NOM 062-ZOO-1999) and approved by

the Bioethics Committee of the School of Natural Sciences of the Universidad Autónoma de Querétaro (approval no.: 45FCN2022).

Following a one-week acclimatization period with ad libitum access to a standard diet (SD; 52% carbohydrates, 22% protein, 6% fat; Laboratory Rodent Diet 5001, Land O'Lakes, MN, USA) and water, rats were randomly assigned to seven experimental groups (*n* = 10) (Figure 1). These groups included one healthy control group fed SD throughout the experiment and one diabetic control group fed a high-fat high-fructose diet (HFFD) throughout the experiment. The HFFD diet that was made with standard rodent feed powdered and 20% porcine fat and 20% fructose (Crystalline fructose, ADM®, Chicago, IL, USA) was added. The HFFD with RBR was prepared by adding 6% of the dry base of the dried and pulverized RBR residue to the HFFD. Both diets were stored at −20 °C until administration.

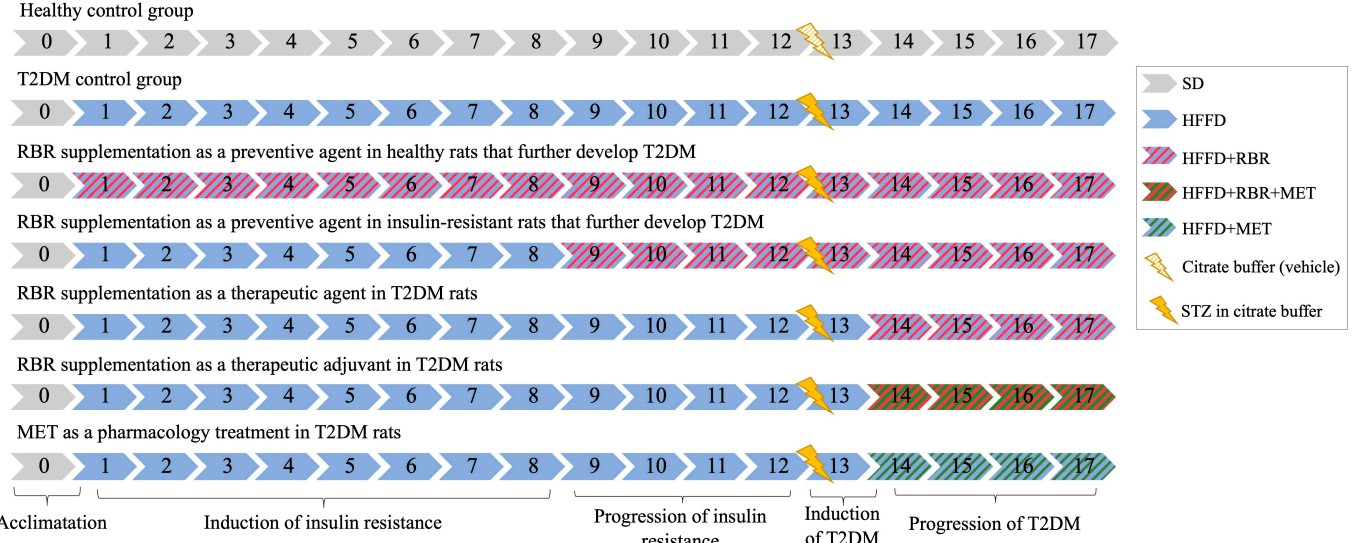

**Figure 1.** Experimental design for the evaluation of roselle beverage residue as a preventive and therapeutic agent in a diabetic rat model. HFFD: high-fat and -fructose diet, MET: metformin, RBR: roselle beverage residue, SD: standard diet, STZ: streptozotocin, TD2M: type 2 diabetes mellitus.

Serum glucose and insulin levels were assessed to monitor diet-induced insulin resistance (IR), and blood samples were collected from the caudal vein of the SD and HFFD control groups. For T2DM induction, HFFD-fed and with RBR-supplemented rats received a single intraperitoneal dose of streptozotocin (30 mg/kg dissolved in a 0.1 M citrate buffer at pH 4.5) at the beginning of week 13. Fasting glucose levels were measured one week later using a digital glucometer (Accu-Check® Instant, Roche Diabetes Care, Indianapolis, IN, USA). Rats with blood glucose levels ≥ 160 mg/dL were considered to have T2DM and continued the HFFD regimen for four weeks. Only citrate buffer was administered by the same route to the ten experimental units corresponding to group (SD).

The experimental strategy included preventive and therapeutic supplementation with RBR starting from different time points of the experimental period (17 weeks), to assess its effects on insulin resistance and T2DM progression. To assess its potential as a preventive nutraceutical agent, RBR was supplemented from week 1 to healthy rats that subsequently developed T2DM and from week 9 to IR rats that subsequently developed T2DM. To assess its therapeutic potential, RBR was supplemented from week 14 to T2DM rats alone or in combination with metformin (MET; 200 mg/kg body weight/day, via intragastric gavage). A positive control diabetic group was treated only with MET.

Daily food and water intake were recorded, and changes in body weight were monitored weekly throughout the study. At the end of the 17th week, rats were fasted for 12 h, anesthetized with sodium pentobarbital (individually dosed via intraperitoneal injection, Sedalpharma®, Pet's Pharma, Mexico city, Mexico), and subsequently euthanized by decap-

itation (Harvard Apparatus, 55-0012, Cambridge, MA, USA) in accordance with established ethical protocols (NOM-194-SSA1-2004 and NOM-033-SAG/ZOO-2014).

Blood samples were collected, centrifuged at $3000 \times g$ at 4 °C for 10 min to separate serum, and then stored at $-70$ °C for subsequent analysis. Liver and skeletal muscle tissues were dissected and rinsed with 0.85% physiological solution, snap-frozen in liquid nitrogen, and stored at $-70$ °C. A small portion of the pancreas was stored in 10% buffered formalin solution (pH 7.6) at room temperature for histological analysis.

### 2.3. Biochemical Analysis

Serum glucose and insulin levels were determined using commercial kits (Spinreact, Girone, Spain; ALPCO Diagnostics, Salem, NH, USA) following the manufacturer's instructions (Selectra Pro M, ELITech Group, Paris, France; Maglumi 600, Snibe, Shenzhen, China). Insulin resistance (IR) was assessed using the homeostatic model assessment index (HOMA-IR), beta cell function index (HOMA-Beta), quantitative insulin-sensitivity check index (QUICKI), and fasting glucose-to-insulin ratio (FGIR), as described by Cacho et al. [16]. Serum urea, creatinine and uric acid levels were determined with commercial kits (Spinreact) following the manufacturer's instructions (Selectra Pro M, ELITech Group). Creatinine and urea values were used to assess the glomerular filtration rate, as proposed by Besseling et al. [17].

### 2.4. Gene Expression Analysis

RNA extraction was carried out from 100 mg of liver and muscle in accordance with Chomczynski and Sacchi [18] using the trizol technique (Trizol™ Invitrogen, Waltham, MA, USA) and following the manufacturer's instructions. Integrity was verified via electrophoresis in 1% agarose gel (UltraPure™, Invitrogen, USA), and purity was quantified with a NanoDrop spectrophotometer using the ratios 260/280 and 260/230 nm (NanoDrop 2000, ThermoScientific, Waltham, MA, USA).

cDNA was synthesized with the reagents (Promega, Madison WI, USA). An amount of 1 μL of whole RNA was used and adjusted to 1000 ng/μL, and 1 μL of Oligo (dT) (0.5 μg/μL) and 13 μL of sterile water were added, subsequently incubated at 70 °C/5 min (BIORAD C1000 CFX96™, Hercules, CA, USA), and then incubated at 4 °C/1 min. An amount of 5 μL of $5\times$ M-MLV reaction buffer, 1 μL dNTP (10 mM), 25 units of ribonuclease inhibitor (0.6 μL), and 200 units of M-MLV RT reverse transcriptase (1 μL) were added and adjusted with sterile water to a total volume of 25 μL. They were incubated at 42 °C/60 min (BIORAD C1000 CFX96™, USA). mRNA expression was determined via real-time PCR (Lightcycler® 96, Roche Diagnostics Co., Indianapolis, IN, USA). An amount of 1 μL of the cDNA was used, and 5 μL of master mix, 0.3 μL of each primer (10 μM) and 3.4 μL of RNase-free water were added. Then, the samples were preincubated at 95 °C for 10 min, and subsequently underwent 40 cycles of denaturation at 95 °C for 15 s, alignment at 64 °C/40 s, and amplification at 72 °C/10 s. The melting curves were obtained with the following gradient: 95 °C for 15 s, 60 °C for 1 min, and 95 °C for 15 s. The expression of the glucose-6-phosphatase catalytic subunit 1 (*G6pc1*) and phosphoenolpyruvate carboxykinase 1 (*Pck1*) in liver was assessed to evaluate gluconeogenesis, whereas that of insulin receptor substrate 1 (*Irs1*) in the skeletal muscle was determined to assess insulin cascade signaling. Relative mRNA expression was calculated via normalization with *Bactin* as the housekeeping gene in accordance with the $2^{-\Delta\Delta Ct}$ method [19]. The following primer sequences were used for *Irs1* (Fw: GACGCTCCAGTGAGGATTTAAG; Rv: GGAGGATTGCTGAGGTCATTTAG), *G6pc1* (Fw: GTGGTTGGAGACTGGTTCAA; Rv: CACGGAGCTGTTGCTGTAATA), and *Pck1* (Fw: GAGCTGTTCGGAATCTCTAAGG, Rv: TCGGAGCTCCCTCTCTATTT).

### 2.5. Histologic Analysis

Pancreas samples fixed in buffered formalin solution were processed for histologic analysis using Hematoxylin and Eosin staining. Langerhans islets were observed under a microscope (Olympus CellSens Entry, Olympus Co., Tokyo, Japan) at $10\times$ and $40\times$ magnification.

*2.6. Statistical Analysis*

Data were analyzed using JMP software v16 (JMP Statistical Discovery LLC, Cary, NC, USA). Extreme values were identified using box and whisker plots (>3.0 IQR) and excluded. The Kolmogorov–Smirnov and Levene tests were used to assess normality and variance homogeneity, respectively, with a significance level set at $p < 0.05$. Parametric variables were analyzed using Tukey's and Dunnett's tests, while non-parametric variables were assessed using the Wilcoxon test ($p < 0.05$ considered statistically significant).

## 3. Results and Discussion

*3.1. Dietary Fiber and Polyphenol Contribution of RBR Supplementation*

In this study, we aimed to assess the potential RBR as a dietary supplement at different stages of T2DM development and progression, including preventive and therapeutic strategies. Based on our previous comprehensive characterization of RBR's nutrimental and nutraceutical properties [14], the RBR dose used in this study yielded 4.9 g of total dietary fiber and 2.6 mg of total polyphenols (comprising both extractable and non-extractable forms), with an associated antioxidant capacity of 27.2 mmol TE (Trolox equivalents) (Table 1). Translating these findings into a human-equivalent dose, following the methodology outlined by Nair and Jacob [20], revealed an approximate dose of 0.9 g of RBR/kg of body weight. For an average 60 kg individual, this corresponds to roughly 12 g of RBR per day, which equates to 10 g of dietary fiber and 0.5 g of total polyphenols. The recommended daily intake of dietary fiber for humans is 25–30 g, while for polyphenols, it is 1 g/day. Specifically, the RBR supplement provides 33–40% of the recommended dietary fiber intake and 50% of the suggested polyphenol intake for humans. Importantly, clinical studies have demonstrated substantial improvements in glycemic control among T2DM patients with a median dietary fiber supplementation dose of 10 g/day [4].

**Table 1.** Nutrient and nutraceutical composition of the roselle beverage residue dietary supplement.

| Component | Daily Dose [1] |
|---|---|
| Nutrient composition | |
| Protein | 0.34 g |
| Fat | 0.02 g |
| Carbohydrates | |
| Total dietary fiber | 4.89 g |
| Soluble dietary fiber | 1.07 g |
| Insoluble dietary fiber | 3.85 g |
| Nutraceutical composition | |
| Extractable polyphenols | 0.92 mg GAE |
| Extractable flavonoids | 0.49 mg CE |
| Extractable anthocyanins | 0.08 mg C3GE |
| Extractable anthocyanidins | 0.05 mg CA |
| Acid hydrolysable polyphenols | 0.81 mg GAE |
| Alkaline hydrolysable polyphenols | 0.66 mg GAE |
| Non-extractable proanthocyanidins | 0.16 mg PE |
| Trolox equivalent antioxidant capacity [2] | |
| Extractable fraction | 24.61 mmol TE |
| Acid hydrolysable fraction | 1.37 mmol TE |
| Alkaline hydrolysable fraction | 1.16 mmol TE |

[1] Dose of equivalent of 1 mg RBR/kg/day [14]. [2] Measured using the ABTS· radical scavenging assay. GAE: gallic acid equivalent; CE: (+)-catechin equivalents; C3GE: cyanidin 3-*O*-glucoside equivalents; PE: proanthocyanidins equivalents; TE: Trolox equivalents.

*3.2. Effect of Roselle Beverage Residue on Glucose Homeostasis in T2DM-Induced Rats*

To induce insulin resistance, rats were subjected to an eight-week HFFD, resulting in a normoglycemic state with slightly elevated insulin levels (1.6-fold) and an increase in the HOMA-IR index (1.6-fold) compared with those of rats fed a standard diet (SD) (data not included). Subsequent continuation of the HFFD for an additional four weeks further

exacerbated insulin resistance, evidenced by a significant rise in insulin levels (1.8-fold) and the HOMA-IR index (2.1-fold). Subsequently, type 2 diabetes mellitus (T2DM) was induced, allowing for four weeks of disease progression. At the end of the 17th week, the T2DM-induced rat model displayed well-established hyperglycemia (2.9-fold) with pronounced hypoinsulinemia (4.4-fold), attributed to reduced Langerhans islet size (Figure 2A,B). Consequently, the HOMA-Beta index was substantially decreased (28.2-fold), indicative of beta-cell dysfunction and impaired insulin secretion for blood glucose metabolism.

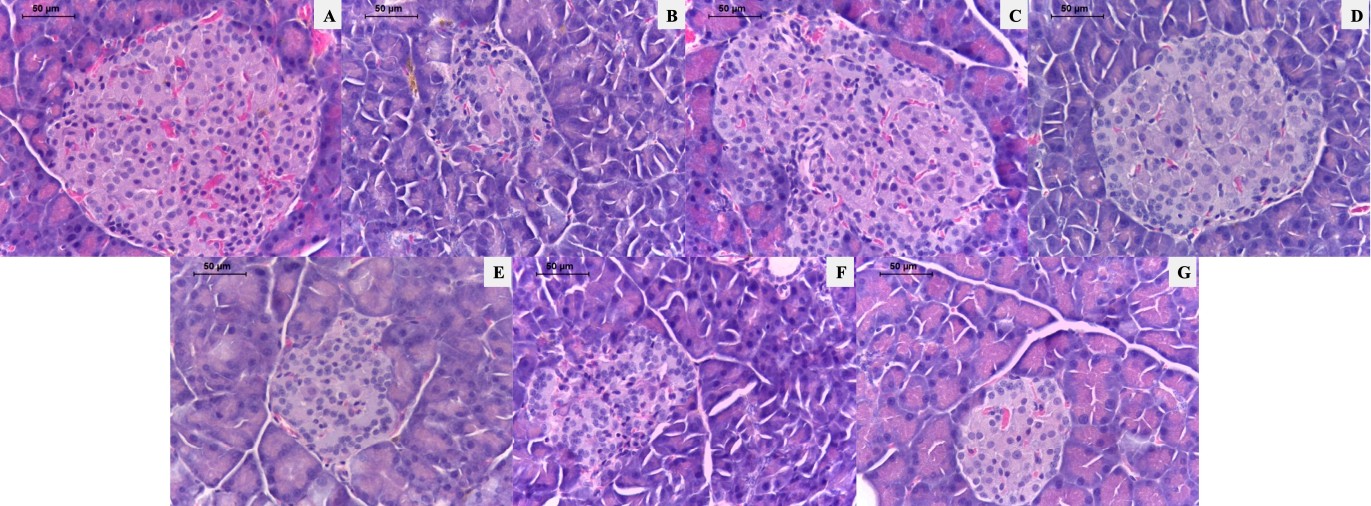

**Figure 2.** Histologic analysis of pancreas of SD-fed rats (**A**), HFFD-fed and STZ-induced rats (**B**), HFFD-fed and STZ-induced rats supplemented with RBR from week 1 (**C**), week 9 (**D**), and week 13 (**E**), HFFD-fed and STZ-induced rats supplemented with RBR and treated with MET from week 13 (**F**), and HFFD-fed and STZ-induced rats treated with MET from week 13 (**G**). HFFD: high fat and fructose diet; MET: metformin; RBR: roselle beverage residue; SD: standard diet; STZ: streptozotocin.

Interestingly, the insulin resistance induced by the HFFD was mitigated post-STZ injection, evidenced by unchanged QUICKI and a slight decrease in the HOMA-IR index, as previously reported by Bonam et al. [21]. Notably, T2DM-induced rats exhibited significantly elevated FGIR values (19.0-fold) compared with healthy rats, due to severe hyperglycemia associated with hypoinsulinemia, as previously reported by Jacobsen et al. [22]. These clinical characteristics collectively denote the establishment of a late-stage T2DM rodent model [23].

RBR supplementation to healthy rats (from week 1) and insulin-resistant rats (from week 9) significantly reduced serum glucose levels 1.6- and 1.8-fold, respectively, compared with those of the T2DM control group (Table 2). This antihyperglycemic effect concurred with increased serum insulin levels, comparable to those in the healthy control group, resulting in a 7.9- and 6.5-fold elevation in the HOMA-beta index. Accordingly, Langerhans islet size increased 1.6–1.7-fold compared with that of the T2DM control group (Table 2, Figure 2C,D).

Moreover, RBR supplementation effectively managed T2DM-induced hyperglycemia when administered to T2DM rats from week 14, resulting in a 1.4-fold reduction in serum glucose levels compared with those in the T2DM control group. However, no significant changes were observed in serum insulin levels or Langerhans islet size (Table 2, Figure 2E,F). Notably, combining RBR supplementation with MET treatment from week 14 resulted in an enhanced hypoglycemic effect (2.0-fold). It is noteworthy that the positive control group administered with MET alone failed to adequately regulate T2DM-induced hyperglycemia and hypoinsulinemia, as its values were statistically like those of the T2DM control group. This finding is similar with reports of monotherapy with metformin inadequately addressing β-cell dysfunction in T2DM individuals [24].

**Table 2.** Effect of roselle beverage residue on serum glucose, serum insulin, insulin sensitivity indices, and Langerhans islet size in diabetic rats.

| Model | SD | HFFD+STZ | | | | | |
|---|---|---|---|---|---|---|---|
| Group | | | Preventive Strategy | | Therapeutic Strategy | | |
| Treatment | Healthy Control | Diabetic Control | RBR from Week 1 | RBR from Week 9 | RBR from Week 14 | RBR+MET from Week 14 | MET from Week 14 |
| Serum glucose (mg/dL) | 152.6 ± 15.3 * | 448.5 ± 36.0 † | 275.0 ± 78.8 * | 242.7 ± 108.4 * | 316.5 ± 99.0 *† | 224.8 ± 83.7 *† | 460.0 ± 25.6 † |
| Serum insulin (ng/mL) | 8.77 ± 3.05 * | 1.98 ± 1.48 † | 6.08 ± 1.69 * | 6.82 ± 3.98 * | 2.18 ± 1.19 † | 3.90 ± 2.76 † | 3.72 ± 2.78 † |
| HOMA-IR | 13.84 ± 5.44 | 6.24 ± 2.63 | 12.82 ± 7.15 | 10.99 ± 6.22 | 9.48 ± 5.49 | 11.89 ± 3.52 | 11.02 ± 6.12 |
| HOMA-Beta | 35.80 ± 12.11 * | 1.27 ± 0.49 † | 10.04 ± 7.74 † | 8.31 ± 7.29 † | 4.19 ± 3.47 † | 19.16 ± 8.86 * | 2.21 ± 1.49 † |
| QUICKI | 0.22 ± 0.01 * | 0.24 ± 0.01 † | 0.22 ± 0.01 * | 0.23 ± 0.01 | 0.23 ± 0.01 | 0.23 ± 0.01 | 0.23 ± 0.01 |
| FGIR | 0.78 ± 0.31 * | 14.79 ± 5.58 † | 1.92 ± 0.89 * | 1.05 ± 0.49 * | 3.13 ± 1.94 * | 1.29 ± 0.38 * | 11.30 ± 3.67 † |
| Langerhans islet size (μm) | 249.0 ± 29.3 * | 132.9 ± 22.4 † | 211.9 ± 20.4 * | 229.6 ± 13.4 * | 164.5 ± 19.9 * | 172.6 ± 10.7 † | 167.7 ± 6.9 † |

Data are expressed as the mean value and standard deviation of ten biological replicates with three technical replicates. * Indicates significant differences ($p < 0.05$) compared with the HFFD+STZ control group. † Indicates statistical differences ($p < 0.05$) compared with the SD control group. FGIR: fasting glucose-to-insulin ratio; HFFD: high-fat and -fructose diet; HOMA-Beta: homeostatic model assessment index for beta cell function; HOMA-IR: homeostatic model assessment index for insulin resistance; MET: metformin; QUICKI: quantitative insulin-sensitivity check index; RBR: roselle beverage residue; SD: standard diet; STZ: streptozotocin.

The beneficial effects of RBR could be attributed to its rich content and diversity of bioactive compounds. For instance, studies have shown that both soluble and insoluble dietary fibers contribute to reducing hyperglycemia in T2DM rodents [25]. In terms of polyphenols, our previous research revealed that RBR contains a substantial amount of extractable caffeoylquinic acids, along with hydroxybenzoic acid and its derivatives in both extractable and non-extractable forms. Furthermore, the deep red color of RBR is primarily due to delphinidin sambubioside, a disaccharide conjugate and the major anthocyanin identified in RBR [14]. The bioactivity of hibiscus acid, the major organic acid found in RBR, has been poorly studied [26], and its effect on diabetes remains unreported.

A meta-analysis conducted by Araki et al. [27] suggested that delphinidin-based anthocyanins are inversely associated with insulin resistance; however, the specific antidiabetic potential of delphinidin sambubioside has yet to be investigated. Nonetheless, delphinidin 3-*O*-glucoside has been identified as an effective insulin secretagogue [28], and similar bioactivity has been reported for 4-hydroxybenzoic acid [29].

Caffeoylquinic acids are known to stimulate the insulin signaling cascade. For example, neochlorogenic acid has been shown to improve insulin resistance in HFFD rats by upregulating the expression of Glut4 in muscle tissue and activating the AKT and AMPK pathways [30]. Similar mechanisms have been observed for chlorogenic acid, which competitively inhibits G6Pase, thereby reducing hepatic glucose production, as well as α-amylase and α-glucosidase, leading to decreased glucose intestinal digestion [31].

### 3.3. Effect of Roselle Beverage Residue on Gene Expression Related to T2DM Onset and Development

To elucidate the molecular mechanisms underlying the effects of RBR supplementation on T2DM onset and development, we conducted relative expression analysis of key genes involved in glucose metabolism in both liver and skeletal muscle tissues (Figure 3). These organs were chosen due to their pivotal roles in maintaining glucose homeostasis. Skeletal muscle accounts for approximately 80% of insulin-dependent postprandial glucose uptake, while the liver contributes around 20% of total glucose production, crucial for sustaining blood glucose levels, particularly during fasting or low glucose availability [32].

Our study revealed alterations in gene expression related to glucose metabolism in the T2DM control group compared with healthy rats; however, no significant differences were observed. Specifically, there was a notable 4.8-fold increase in *Irs1* expression in skeletal muscle, alongside decreases of 15.4- and 2.2-fold in *G6pc1* and *Pck1* expression, respectively, in the liver (Figure 3). While these findings initially appear contradictory to the dysregulated glucose metabolism observed in T2DM, they underscore the complex adaptive responses occurring in different tissues in response to insulin deficiency.

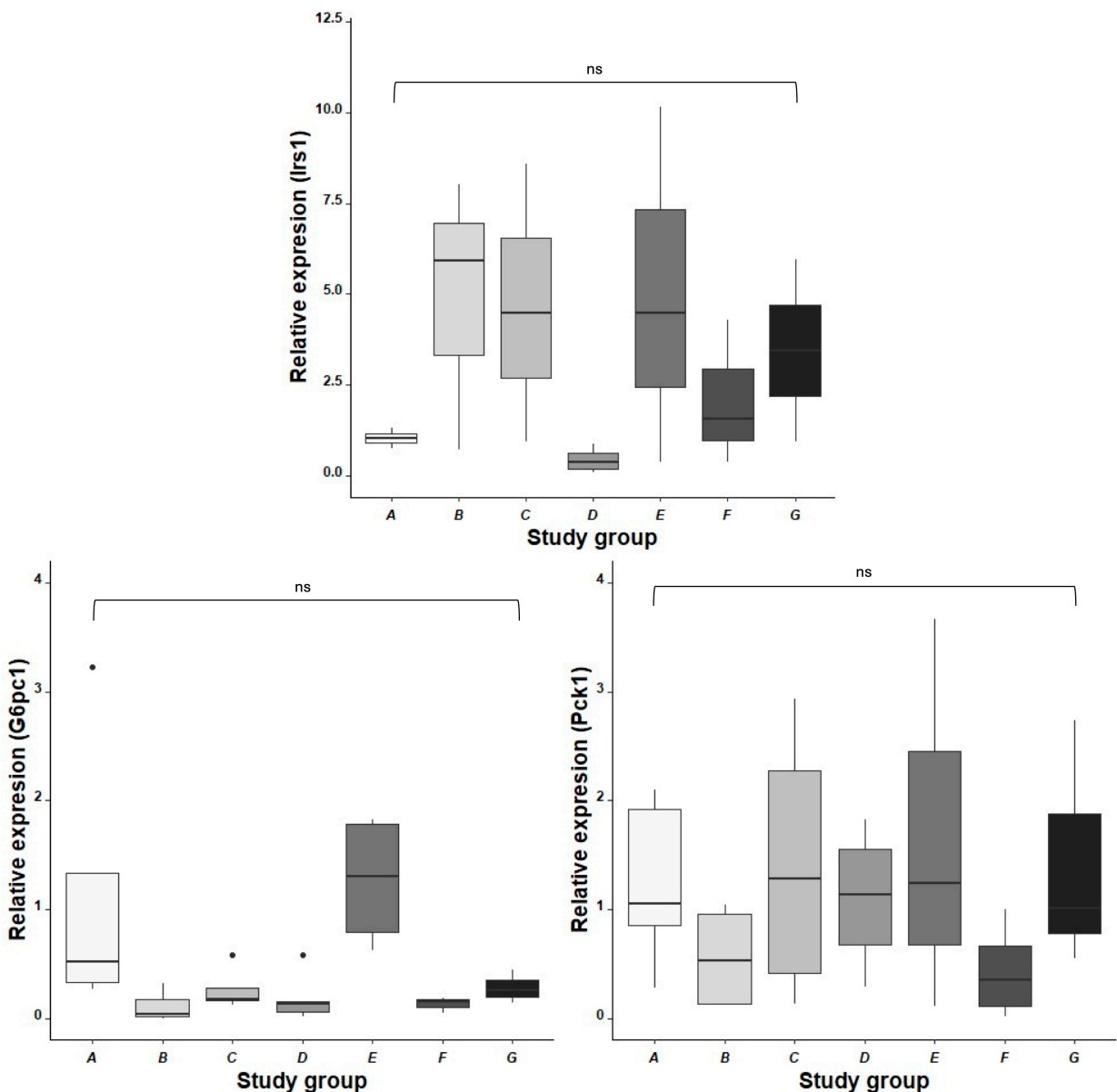

**Figure 3.** Effect of roselle beverage residue on the modulation of key genes involved in glucose metabolism in the liver and skeletal muscle of diabetic rats. (A) SD-fed rats; (B) HFFD-fed rats and STZ-induced rats; (C) HFFD-fed and STZ-induced rats supplemented with RBR from week 1, from week 9 (D) and (E) from week 13; (F) HFFD-fed and STZ-induced rats supplemented with RBR and treated with MET from week 13; and (G) HFFD-fed and STZ-induced rats treated with MET from week 13. [ns] No significant differences ($p < 0.05$) were observed between the experimental groups. '·' Extreme values. G6pc1: glucose-6-phsphatase catalytic subunit 1; HFFD: high-fat and -fructose diet; Irs1: insulin receptor substrate 1; MET: metformin; Pck1: phosphoenolpyruvate carboxykinase 1; RBR: roselle beverage residue; SD: standard diet; STZ: streptozotocin.

Vorotnikov et al. [32] hypothesized that impairments associated with insulin resistance may manifest primarily in the distal part of the insulin signaling cascade, rather than the proximal part typically examined. These results are similar with our observations, indicating that alterations in gene expression, particularly in skeletal muscle, may represent compensatory mechanisms aimed at enhancing insulin sensitivity in the context of insulin deficiency or as a consequence of a distal response.

Contrary to common observations of the increased expression of gluconeogenic genes in T2DM, Samuel et al. [33] reported unaltered gluconeogenic gene expression in a rat

model of poorly controlled T2DM induced by STZ in combination with a high-fat diet. This challenges the notion that PEPCL and G6Pase play pivotal roles in augmented hepatic glucose production in T2DM. While our results suggest a complex interplay between insulin sensitivity and glucose metabolism, further research is warranted to elucidate the precise mechanisms underlying these changes and their contribution to T2DM progression. While some RBR-supplemented groups demonstrated improvements in hyperglycemia, including increased insulin secretion in preventive nutraceutical strategies, the precise mechanisms driving these effects require further investigation.

### 3.4. Effect of Roselle Beverage Residue on Kidney Function in T2DM-Induced Rats

In this study, we assessed the effect of the RBR on renal function, since nephropathy is a common complication associated with poorly controlled T2DM. The HFFD+STZ control group showed unaltered serum urea and creatinine levels as compared with the SD control group (Table 3). However, the glomerular filtration rate significantly increased 1.3-fold, indicating glomerular hyperfiltration. According to the classic course of diabetic kidney disease, glomerular hyperfiltration is observed as a consequence of hyperglycemia-induced structural and dynamic alterations, preceding the progression of nephropathy [34].

**Table 3.** Effect of roselle beverage residue on kidney function biomarkers in diabetic rats.

| Model | SD | | HFFD+STZ | | | | |
|---|---|---|---|---|---|---|---|
| Group | | | Preventive Strategy | | Therapeutic Strategy | | |
| Treatment | Healthy Control | Diabetic Control | RBR from Week 1 | RBR from Week 9 | RBR from Week 14 | RBR+MET from Week 14 | MET from Week 14 |
| Serum urea (mg/dL) | $43.00 \pm 1.42$ | $33.67 \pm 3.65$ | $34.50 \pm 2.03$ | $36.38 \pm 3.50$ | $34.11 \pm 3.10$ | $28.33 \pm 3.76$ | $34.33 \pm 2.88$ |
| Serum Creatinine (mg/dL) | $0.36 \pm 0.02$ | $0.30 \pm 0.03$ | $0.31 \pm 0.01$ | $0.35 \pm 0.02$ | $0.31 \pm 0.01$ | $0.33 \pm 0.02$ | $0.31 \pm 0.01$ |
| Serum uric acid (mg/dL) | $0.52 \pm 0.07$ * | $1.23 \pm 0.25$ [†] | $0.59 \pm 0.09$ * | $0.56 \pm 0.09$ * | $0.70 \pm 0.10$ | $0.69 \pm 0.09$ | $0.83 \pm 0.14$ |
| Glomerular filtration rate (μL/min) | $3866 \pm 99$ * | $5042 \pm 377$ [†] | $4159 \pm 211$ | $4098 \pm 99$ | $4627 \pm 323$ | $4627 \pm 323$ | $4318 \pm 157$ |

Data are expressed as the mean value and standard deviation of ten biological replicates with three technical replicates. * Indicates significant differences ($p < 0.05$) compared with the HFFD+STZ control group. [†] Indicates statistical differences ($p < 0.05$) compared with the SD control group. HFFD: high-fat and -fructose diet; MET: metformin; RBR: roselle beverage residue; SD: standard diet; STZ: streptozotocin.

It has been extensively reported that dietary fructose is closely related with hyperuricemia, which is characterized by an imbalance between uric acid production and kidney clearance, leading to its accumulation [35]. It has been proposed that uric acid promotes vascular injury via inflammation and/or the activation of the renin-angiotensin-aldosterone system, promoting renal arteriolar damage and the progression of diabetic kidney disease [36]. Therefore, the rodent model obtained in this study showed clinical traits of late-stage T2DM with early renal dysfunction.

According to the ADA, glucose management is required to reduce the risk of developing diabetic nephropathy, or to slow its progression [37]. In this regard, in our study, supplementation with RBR in both preventive and therapeutic strategies reduced serum glucose levels compared with those of the HFFD+STZ control group (Table 2); however, no beneficial effect was observed on the glomerular filtration rate (Table 3). Nevertheless, rats supplemented with RBR prior the development of T2DM showed significantly lower serum uric acid levels as compared with the HFFD+STZ control group (2.1–2.2-fold).

Interestingly, the antihyperuricemic activity of roselle ethanol extracts has been previously reported [38]. Zhou et al. [39] demonstrated that chlorogenic acid, a major polyphenol of RBR, reduces the activity of hepatic xanthine oxidase (XOD) and reduces the expression of renal organic anion transporter 1 (OAT1) in hyperuricemic mice, whereas Xie et al. [40] proposed delphinidin 3-*O*-sambubioside as a XOD inhibitor. These results indicate that polyphenols can be proposed as pharmacologic agents against hyperuricemia, since XOD is

the main rate-limiting enzyme involved in uric acid biosynthesis, whereas OATs reabsorb about 90% of filtered uric acid in the proximal convoluted tubule [41].

## 4. Conclusions

Our study demonstrates that supplementation with RBR has beneficial effects in preventing and managing late-stage T2DM in rodents, with greater efficacy observed when administered in both healthy and insulin-resistant subjects. These effects include improvements in insulin sensitivity, reductions in serum glucose levels, and enhancements in pancreatic function. However, changes in gene expression, notably in Irs1, G6pc1, and Pck1, raise questions about the underlying mechanisms of RBR on glucose metabolism. Despite not observing a direct renoprotective effect, our findings demonstrate the preventive role of RBR in mitigating hyperuricemia, a metabolic alteration associated with the progression of diabetic nephropathy. These findings highlight the multifaceted effects of RBR beyond glycemic control and insulin sensitivity; nonetheless, further investigations are warranted to fully elucidate the mechanisms underlying these beneficial effects. While our findings support the use of RBR as a promising adjunctive therapy for T2DM, further research, including human clinical trials, is needed to validate its effectiveness and elucidate its potential therapeutic benefits.

**Author Contributions:** Conceptualization, M.A.A.-L. and I.F.P.-R.; formal analysis, E.R.-R., J.E.S.-T., D.G.G.-G., R.R.-C., O.P.G. and I.F.P.-R.; funding acquisition, I.F.P.-R.; investigation, E.R.-R. and J.E.S.-T.; resources, D.G.G.-G., R.R.-C., M.A.A.-L. and I.F.P.-R.; supervision, M.A.A.-L. and I.F.P.-R.; writing—review and editing, E.R.-R., J.E.S.-T., D.G.G.-G., R.R.-C., O.P.G. and M.A.A.-L.; writing—original draft, I.F.P.-R. All authors have read and agreed to the published version of the manuscript.

**Funding:** This research was funded by Fondo para el Desarrollo del Conocimiento (FONDEC-UAQ) 2022, grant number FQU-2022-08.

**Institutional Review Board Statement:** Not applicable.

**Informed Consent Statement:** Not applicable.

**Data Availability Statement:** The original contributions presented in the study are included in the article, further inquiries can be directed to the corresponding author.

**Acknowledgments:** The authors are grateful to CONAHCyT for the master's and postdoctoral degree granted, respectively, to J.E.S.-T. and E.R.-R. We thank Ericka Alejandra de los Ríos Arellano for the histology analysis carried out in the Microscopy Unit of the Institute of Neurobiology of the National Autonomous University of Mexico (INB-UNAM).

**Conflicts of Interest:** The authors declare no conflicts of interest.

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
