# Peer review of "Preventive and Therapeutic Efficacy of Roselle Beverage Residue in Late-Stage Type 2 Diabetic Rats"

_beverages, doi:10.3390/beverages10020040_

Round 1
Reviewer 1 Report
Comments and Suggestions for Authors
Attached is the file with the correction suggestions.

Author Response
Thank you for your comments and suggestions to improve the quality of our paper. Please find the point-by-point responses attached and the corresponding revisions highlighted in red in the re-submitted files.

Reviewer 2 Report
Comments and Suggestions for Authors
The present study investigated the potential of Roselle beverage residue (RBR) as a preventive and therapeutic agent for type 2 diabetes mellitus (T2DM) in a rat model fed with a high-fat high-fructose diet (HFFD) for 17 weeks, which reaching insulin resistance by week 9, and induced to T2DM with streptozotocin (STZ) at week 13. The results showed that all preventive and therapeutic RBR-supplemented groups exhibited reduced serum glucose levels (1.4-fold to 1.8-fold) compared to the HFFD+STZ control group. Preventive RBR administration enhanced pancreatic function, leading to improved insulin sensitivity (6.5-fold to 7.9-fold). Gene expression analysis of key genes involved in glucose metabolism in liver and skeletal muscle of diabetic rats revealed slight alterations, but no significant differences were observed between the experimental groups. Additionally, RBR supplementation demonstrated a preventive role in mitigating hyperuricemia (2.1-fold to 2.2-fold), with no effect on glomerular hyperfiltration. The exact mechanisms underlying RBR effects remain to be fully elucidated, There have some concerns as listed in the following:
(1) As stated in the Introduction part (L,72-73: specifically, this study aims to assess the impact of RBR on the expression of genes involved in insulin resistance in T2DM-induced rats.), but the present data showed no conclusive results.
(2) Typos and others:
*L76: Hibiscus sabdariffa L. -> Hibiscus sabdariffa (italic letter) L.
*L162: min vs. minutes
*L207: PA?: -> PE:
**L211: a normoglycemic state with slightly elevated insulin levels (1.6-fold) and an increase in the HOMA-IR index (1.6-fold) compared to rats fed a standard diet (SD) (Table 2). -> no such data in Table 2?
L234: as previously by Bonam et al.-> as previously reported by Bonam et al.
** L295: What is the meaning of marked * in the G6pc1 expression picture of Fig. 3
**L309: alongside decreases of 15.4? and 2.2-fold in G6pc1 and Pck1 expression,
**L327: rats supplemented with RBR after T2DM onset exhibited elevated? Irs1 -> not matched with the data in Fig. 3.
*L349: 1mg/dL, 2mL/min.
L453: 162, 156-9-> 162, 156-159
L496: J.Am Soc. Nephrol. -> J. Am Soc. Nephrol.
L507: 278-80-> 278-280
Author Response

(The authors gave the same response as above.)

Reviewer 3 Report
Comments and Suggestions for Authors
The manuscript entitled "Preventive and therapeutic efficacy of roselle beverage in late-stage type 2 diabetic rats" aims to explore the efficacy of roselle beverage residue on the prevention and treatment of type 2 diabetes using rats as animal models, as well as investigate the underlying mechanisms behind this processing residue. The research topic is interesting and fulfills a gap in the research field. The manuscript is well-written and organized in proper sections. Some questions/comments are stated below and should be addressed to clarify some aspects:
1. Title: The work developed is in roselle beverage residue, and not in the roselle beverage. Please revise the title considering this.
2. Authors and affiliations: Please indicate the meaning of "†" after the affiliations.
3. Abstract: Please clarify in the abstract that RBR was administered to animals by gastric gavage. It is not clear in the present form.
4. Keywords: Please format "Hibiscus sabdariffa" in italic letter. Revise it along the manuscript (for example, line 76).
5. Please consider including a list of the abbreviations used throughout the manuscript before the introduction.
6. Introduction: In the last paragraph, consider highlighting the novelty and relevance of the study.
7. Correct "Mexico" throughout the manuscript (in lines 77 and 86, for example).
8. Correct the celsius degrees symbol throughout the manuscript ("ºC").
9. Format "ad libitum" in italic letter (line 92).
10. In line 99, please replace "roselle beverage residue" by the abbreviation "RBR" already defined above.
11. In line 122, add a space between "body" and "weight".
12. Please format "p" from probability value in italic letter (line 185). Revise it along the manuscript.
13. Table 1: Please indicate the reference(s) of the content previously published and stated them in the table caption or directly in the table.
14. Tables 2 and 3: Please indicate the units of the results directly in the table (instead of in the table footnotes).
15. Results and Discussion clearly states the main findings of the study and provides explanations based on the existing literature.
16. Conclusion summarizes the main findings and highlights future perspectives on the research topic.
Author Response

(The authors gave the same response as above.)
